# Comparison of Mechanical Properties of Ductile/Brittle Epoxy Resin BFRP-AL Joints under Different Immersion Solutions

**DOI:** 10.3390/polym15193892

**Published:** 2023-09-26

**Authors:** Haichao Liu, Ziyang Ding, Yisa Fan, Yang Luo, Yang Yang

**Affiliations:** 1School of Mechanical Engineering, North China University of Water Resources and Electric Power, Zhengzhou 450045, China; 19138123694@163.com (Z.D.); fanyisa123@163.com (Y.F.); 2Department of Mechanical Engineering, Faculty of Engineering, University of Malaya, Kuala Lumpur 50603, Malaysia; luoyang2925@163.com; 3Institute of Mechanical Engineering, Materials and Transportation, Peter the Great Saint-Petersburg Polytechnic University, Saint Petersburg 195251, Russia; y691483@163.com

**Keywords:** basalt fiber, FRP (fiber-reinforced polymer), adhesives joint, dipping solution, failure strength

## Abstract

The bonding properties of BFRP composites have been demonstrated in previous studies, satisfying the strength and durability criteria. In this paper, a further in-depth study is carried out to bond Basalt Fibre Reinforced Polymer (BFRP) to Aluminum Alloy 5052 using two bonding agents, Aralite^®^ 2012 and Aralite^®^ 2015, respectively. The salt sprays under 80 °C, 3.5% NaCl environment; 80 °C, 5% NaCl environment; and pure water environment are also considered for comparison. Experimental results show that joints created with Araldite^®^ 2012 adhesives show higher average breaking strength (10.66 MPa at 720 h) and better ductility in a 5% NaCl environment. While the Araldite^®^ 2015 adhesive joint exhibits a combination of tear failure and interface failure, along with thin-layer cohesion failure. In the SEM images of the two adhesive joints’ failure, fiber pullout due to tension and damage at the interface between fiber and resin is apparent. To validate the experimental outcomes, water absorption testing, DSC, TGA-DTG, and FTIR experiments were conducted on dog-bone-shaped adhesive specimens to elucidate the results.

## 1. Introduction

Due to the higher energy efficiency and smaller cost operating requirements of aerospace, marine, automotive, and railway industries, the demand for lightweight components is increasing, and the adoption of lighter alloys, composites, and other materials of lower quality is becoming a trend [1]. In aviation equipment, the most advanced civil passenger aircraft Airbus A350XWB and Boeing 787, composite materials account for more than 50% of the total structure, the application structure covers the bulkhead skin, wing, keel beam, flat tail vertical tail, and other key parts. Each part of the composite material adopts a variety of mutual coordination, so it can be regarded not only as “material” but also as “structure”, providing a broad space for designability from micro to macro, thus providing possibilities for the integration of material structure and design and manufacturing of aviation equipment. It is considered one of the effective ways for the future aviation structure to be lightweight, functional, and intelligent [2].

Composite materials are the most widely needed high-performance engineering materials in aerospace. They have excellent performance in weight reduction and damage tolerance, achieve the purpose of reducing fuel consumption rate and reducing cost, and then gradually replace traditional alloys in the aerospace field [3]. Composites are prepared by embedding or reinforcing two or more external composites in a common matrix. The synergistic effect of mixing two or more materials provides new and superior properties in the material, such as reduced mass and improved elastic modulus, ductility, and flame retardancy [4,5,6]. For polymer-based composites (PMCS), especially fiber-reinforced polymers can replace metal components of aircraft due to their low density, excellent strength and stiffness, and corrosion resistance, thereby reducing aircraft weight and fuel consumption, improving aircraft flexibility and speed [7]. The most widely used reinforcement fibers are carbon fiber, glass fiber, and aryl fiber. In recent years, basalt fiber has also been rapidly developed and used. The advantages of basalt are (I). Basalt is widely found in nature and makes up 90% of all volcanic rocks; (II). Basalt fibers are also superior to carbon fibers and glass fibers in terms of quality and cost; (III). Basalt fibers also exhibit good endurance properties at high temperatures, such as mechanical properties, low water absorption, and acid resistance [8,9,10]. Wang et al. [11] studied the mechanical properties of polypropylene (PP) and short basalt fiber-reinforced polypropylene (BFRPP) under different strains, where BFRPP is made of polypropylene (PP) matrix and basalt fiber-reinforced material. The tensile strength and Young’s modulus of BFRPP material are better than those of PP material at strain rates of 0.001 s^−1^–400 s^−1^, and the maximum tensile strength of BFRPP material is 71.7 MPa. The 5052 aluminum alloy belongs to AL-MG alloy, with a wide range of uses, good corrosion resistance, excellent welding, good cold working, and medium strength. Liu et al. [12] studied the fatigue reliability of the self-piercing rivet joints of a 5052 aluminum alloy plate. By combining the test results with the simulation of the connection process, the static and fatigue properties of SPR nodes are tested and analyzed. According to the corresponding P-S-N curves, the minimum fatigue limit of SPR nodes under different survival probabilities and given confidence can be calculated, which is of great significance for the reliability design of SPR structures.

Although the composite material has gradually become the main load-bearing structure, it cannot completely replace the metal material at present due to its complex mechanical properties, so there is inevitably a structure connecting the composite material and the metal material. Mechanical fastening [13,14] and adhesive bonding [15,16] are currently the two main ways of joining aerospace components. Conventional fasteners (bolt and rivet joints) usually result in fiber cutting, which causes stress concentration. In contrast, the use of adhesive joints allows efficient connection of large thin-walled components with better continuity to evenly distribute the load, and they also offer advantages such as a high strength-to-weight ratio, design flexibility, and ease of fabrication [17]. Adhesives require high strength to effectively bond to load-bearing structures. The epoxy resin adhesive is a kind of structural adhesive that can produce good bonds with various materials and has chemical corrosion resistance and high thermal stability [18]. The epoxy resin-based adhesive is composed of an epoxy resin prepolymer and curing agent. During the curing reaction of components, a three-dimensional polymer network is formed [19]. Qin et al. [20] used Araldite^®^ 2015 adhesive to bond CFRP and aluminum alloy with shear joints (TSJs), 45° inclined joints (SJ45), and butt joints (BSJs). After environmental aging, the failure strength of TSJs decreased from 26 MPa to 23 MPa after 30 days of aging. The failure strength of the other two joints decreases more obviously, but the minimum failure strength of all joints is still around 20 MPa after aging, which still has high bond strength.

Adhesive joints are often exposed to a variety of complex environments, and their environmental durability depends on the resistance generated by the internal structure of each component, as well as on the combination between them [21]. Hygrothermal Aging is the coupling effect of moisture and temperature, which is one of the most severe exposure conditions known. The diffusion rate of water in the structural adhesive increases significantly with the rising temperature and accelerates into the bonding area of the joint. Moisture can diffuse through the bulk adhesive, transport along the interface, capillary action, and diffusion into the adhesive matrix through porous adherent, and it also creates (I). Changing the resin matrix; (II). Disruption of the fiber/matrix interface; (III). Fiber grade degradation [22,23]. Li et al. [24] summarized the effects of the tensile properties of GFRP/BFRP composites under seawater aging, SWSSC, and strain rate. The degradation degree of FRP composites is serious with the increase in soaking time, temperature, stress level, and solution alkalinity/salinity. The mechanical properties and failure mode of FRP composites are affected by strain rate.

The high-performance composite material has the characteristics of high material structure integration due to its microstructure. High-performance composites have a high degree of material-structure integration due to their microstructures. Based on the actual conditions of aviation service and satisfying the use in complex environments, a suitable cementing agent and cooperation scheme are proposed to achieve the integration of joint design, material selection, and performance evaluation. Under the previous research results on the bonding of BFRP substrates, this study focuses on the bonding between composites and metals [25]. In this paper, BFRP-TI joints were used to compare the tensile strength of epoxy adhesive Araldite^®^ 2012 and Araldite^®^ 2015 joints at 80 °C/DI water, 80 °C/3.5% NaCl solutions, and 5% NaCl solutions, and DSC, FTIR, and TGA-DTG experiments were carried out to explain the mechanical properties of the bonded joints.

## 2. Experimental Process

### 2.1. Material Selection

Basalt fiber reinforced resin composite—aluminum alloy (BFRP-Al) joint is based on a BFRP composite sheet and 5052 aluminum alloy. The thickness of the BFRP composite sheet (Zhongdao Technology Company, Changchun, Jilin Province, China) is 2 mm, it is made of twill and unidirectional prepreg, the direction of fiber layer spreading is [0/90/0/90/0/90], single fiber diameter is 11 μm, the density is 600 g/cm^3^. BFRP composite resin (Shanghai Huibo New Material Technology Co., Ltd., Shanghai, China) consists of GT-807A (epoxy resin) and GT-807B (hardener), and the composition ratio (mass ratio) is 100:20. The volume yarn content of BFRP plate was 65.18%. The 5052 aluminum alloy is a high-quality aluminum alloy produced by heat treatment and pre-stretching process, with a thickness of 2 mm. The Araldite^®^ 2012 and Araldite^®^ 2015 epoxy resin adhesives manufactured by Huntsman Company in the United States were selected. Araldite^®^ 2012 adhesive is a fast, versatile, two-component (the ratio of epoxy resin and the curing agent was 1:1), room temperature curing, high viscosity liquid adhesive with high strength and good toughness. Araldite^®^ 2015 is a two-component (the ratio of epoxy resin and curing agent was 1:1), room temperature curing, elastically bonded paste adhesive with very high lapping shear and peel strength and good resistance to dynamic loads. The specific parameters of BFRP composite board, 5052 aluminum alloy, and Araldite^®^ 2012 and Araldite^®^ 2015 adhesives are provided by the manufacturer, as shown in Table 1, Table 2 and Table 3.

### 2.2. Preparation of Test Specimen

Single-lap joints are selected to study the aging law of BFRP-Al joints, and the production standard of bonded joints refers to ISO4587: 2003 [26]. The environment for sample preparation is dust-free, room temperature 25 ± 3 °C, and relative humidity 50 ± 5%. To prevent other factors from affecting the experiment, all test pieces adopt a unified, standard process bonding process. The surface of the 5052 Al base material is first treated with alumina sandblasting to increase the roughness. Sandblasting will cause BFRP to be damaged on the surface resin, so BFRP will not be sandblasted. Then acetone is used to clean the bonding surface of aluminum alloy and BFRP composite material to remove dust and grease. About 15–20 min after the wiping is completed, and after drying to ensure that the adhesive is evenly mixed, a two-component glue gun is used for sizing the substrate. It is required to ensure that a single stress is received during the joint test, so a gasket that is 0.1 mm thicker than the substrate is used. Then the bonding of the test piece on the special fixture is completed, and the specimen fabrication process is shown in Figure 1. A layer of PTFE on the inner surface of the fixture is spread to prevent the excess adhesive from bonding the joint to the fixture. After the bonding is completed, the test piece is cured at room temperature for 120 h and then removed from the fixture, and the remaining glue is cut off. The prepared test piece is put into the high- and low-temperature damp heat alternating experiment box (WS-1000, Weiss Equipment Experiment Company, Taichang, Jiangsu Province, China), and 80 °C/deionized water, 80 °C/3.5% NaCl solutions, and 80 °C/5% NaCl solutions aging tests are completed.

### 2.3. Experimental Scheme

#### 2.3.1. Experimental Design

To study the aging behavior of BFRP-Al joint in 80 °C/deionized water, 80 °C/3.5% NaCl solutions (average concentration in the ocean), and 80 °C/5% NaCl solutions (salt-rich environment) environment, 0 (Not aging), 240, 480, 720 h aging time points were selected. According to different aging time points in three environments, they were divided into 4 groups, 12 groups in total, and 3 specimens were selected from each group. After the specimens are cured, they are put into the high-low temperature damp-heat alternating experiment box and aged to the above time points in the corresponding environment, and high-low temperature damp-heat alternating experiment box is shown in Figure 2 It is taken out after reaching the corresponding time, and the remaining strength of the bonded joint is tested after placing it for 8 h.

#### 2.3.2. Quasi-Static Strength Test

The tensile test of the BFRP-Al single lap joint is carried out based on the standard ASTM D5868-01 [27]. The experimental equipment is the Xinguang Universal Testing Machine (WDW-100A, China Jinan Xinguang Testing Machine Manufacturing Co., Ltd., Jinan, Shangdong Province, China). To eliminate the influence of the bending stress in the stretching process in the experiment, 2 mm thick spacers at both ends of the specimen are installed and stretched at a constant speed of 1 mm/min. The force-displacement curve obtained during the stretching process is recorded by the computer system connected to the universal testing machine. The quasi-static tensile test is shown in Figure 3.

#### 2.3.3. Water Absorption Test

To obtain the variation law of Araldite^®^ 2012 and Araldite^®^ 2015 adhesives under temperature and humidity conditions, the mold of the test piece is designed according to NF ISO 527-2-2012 standard, and the dog bone test piece is made [28]. Dog-bone tensile mold and specimens are shown in Figure 4a,b. Research via the high-precision analysis of the scales (balance accuracy of 0.0001 g) on the adhesive hygroscopicity study and the schematic diagram of analytical balance is shown in Figure 4c. BFRP and adhesive specimens are taken out of the environment chamber every 24 h for measurements. In the process of testing the quality of specimens, disposable dust-free rubber gloves are worn to prevent contamination of specimens, and the surface is gently wiped with absorbent paper to remove moisture. They were grouped according to three different environments, and each group included three samples. The original mass of dog-bone tensile specimens and BFRP made of two kinds of adhesives and the mass at different time points were recorded, respectively.

#### 2.3.4. Glass Transition Temperature Test

A differential scanning calorimeter (DSC Q100, TA Instruments, New Castle, DE, USA) was used to analyze the adhesive before and after aging, and a standard aluminum crucible with the aluminum cap was used for testing in a nitrogen environment. For the experimental samples, a two-step process was carried out. The first-time heating was from −50 °C to 200 °C and holding at 200 °C for 2 min, to eliminate the thermal history that has a large influence on the Tg of the sample. The second cooling was from 200 °C to −50 °C and kept at −50 °C for 2 min; the Tg of the sample was measured. According to GB/T19466.1-2004 [29], Tg can be determined as the intersection of a line with a curve at an equal distance from two extrapolated lines.

#### 2.3.5. Fourier Transform Infrared Spectroscopy

Samples were extracted from the bonded region, and the sample surface was analyzed using a Fourier transform infrared spectrometer (VERTEX 70V, Bruker, Bremen, Germany). The spectra were obtained by total attenuation multiple reflectance (IRATR), and 128 scans were performed. Before analysis, the spectrometer platform was cleaned with dry nitrogen. A total of 128 scans were performed with a spectral range of 400–4000 cm^−1^ and a resolution of 4 cm^−1^.

## 3. Results and Discussion

### 3.1. Moisture Absorption Analysis of Dog-Bone Tensile Sample

Various external conditions, such as high and low temperature, dryness, or long-term water immersion often have a specific impact on the bonding structure, thereby changing the strength of the bonding structure. These negative effects reduce the service life of the bonded structure. The dumbbell sample was used for the water absorption experiment, the data were simulated according to Fick’s second law, and the formula is as follows:(1)$$\frac{M_{t}-M_{0}}{M_{\infty }-M_{0}}=1-\frac{8}{\pi^{2}}\sum_{n=0}^{\infty }\frac{1}{{\left(2n+1\right)}^{2}}\exp\left[-\frac{{\left(2n+1\right)}^{2}\pi^{2}D}{h^{2}}t\right]$$

$M_{0}$ is the initial moisture content in the material, the corresponding value is 0, $M_{\infty }$ is the saturated moisture content, $M_{t}$ is the water content at time *t*. D is the diffusion coefficient, and it can be obtained from the following equation:(2)$$D=\frac{\pi }{16}{\left(\frac{h}{M_{\infty }}\right)}^{2}\left[\frac{M(t_{1})-M(t_{2})}{\sqrt{t_{1}}-\sqrt{t_{2}}}\right]$$

According to Fick’s theory and experimental data of specimen moisture absorption, a fitting curve is drawn to obtain the relationship between the moisture absorption of the specimens and the square root of the aging time. The specimen moisture absorption data and Fick fitting curve are shown in Figure 5. The diffusivity D of the specimen in the temperature–humidity coupled environment was calculated based on the fitting curve. The diffusivity D of adhesive and BFRP and saturated water absorption are shown in Table 4 and Table 5, respectively.

Figure 5a,b show the moisture absorption and fitting curves of the two adhesives under three different environments. According to the image, it can be seen that the moisture absorption behavior of the sample is roughly combined with Fick’s law of diffusion. The moisture absorption curve in an aging environment can be divided into two stages: the initial moisture absorption increases rapidly, but then the rate gradually decreases and finally reaches dynamic. The diffusion of water in adhesives is dominated by filling pores and cavities with water, accompanied by the reaction of adsorbed water molecules with some hydrophilic functional groups [30]. The moisture absorption of dog-bone tensile specimens of Araldite^®^ 2015 adhesive is higher than that of Araldite^®^ 2012, when water penetrates the adhesive; the free water will occupy the vacant position of the adhesive, and the bound water will form multiple hydrogen bonds with the polymer chain, resulting in the reduction of strength [31]. This explains why the deterioration of mechanical properties of Araldite^®^ 2015 adhesive joints after aging is higher than that of Araldit^®^ 2012 adhesive joints. Another important point to note is that water diffusion is slower in bulk adhesives than at the joint interface. In adhesives, due to their surface topology and resin polarity, adhesives exhibit high moisture absorption [32,33]. In contrast, BFRP plates contain high content of basalt with little or no water absorption characteristics; moreover, the diffusion of water in fiber-reinforced materials only exists in the micro gaps between polymer chains, in the cracks and pores between matrix and interface, and in the microcracks in the matrix. Therefore, the moisture absorption of the BFRP plate is much lower than that of the adhesive.

The water absorption of the joint is minimally affected by the concentration of NaCl solution. The hygroscopicity of the joint in 3.5% NaCl solution is stronger than that in 5.0% NaCl solution. In a deionized water environment, when water is inhaled into the adhesive, it will form an electrolyte with water-soluble substances in the adhesive, and under the action of osmotic pressure, water will be driven into the adhesive to balance the concentration difference. However, in the salt solution environment, NaCl solution has a higher concentration of ions, forming reverse osmosis. Water diffuses from the binder and balances the concentration difference between the interior and the environment, resulting in the water absorption of the binder decreasing with the increase in the concentration of salt ions.

Zanni-deffarges and Shanahan [34] compared the diffusion rates of bulk adhesive samples with those of bonded joints and concluded that the diffusion rates of bonded joints were higher. They attributed their observations to the phenomenon of capillary diffusion, in which the higher surface of the dry adhesive causes water to adsorb along the interface.

### 3.2. Differential Scanning Calorimetry Analysis

As the weak part of the joint, the adhesive layer is easily affected by the aging environment, so it is necessary to carry out a thermal analysis of the adhesive before and after aging [21]. Glass transition temperature is an important parameter to determine the service temperature of adhesive and BFRP composites. Figure 6 shows the DSC curves of Araldite^®^ 2012 in three environments.

Table 6 shows the figure of adhesives’ Tg. The dashed line in the figure represents the Tg reference line of the unaged sample. Tg is determined by analyzing the change of function curve between material heat flow and temperature obtained from DSC; the method of determining the Tg of adhesive is similar to [35]. 

The Tg of the Araldite^®^ 2012 and Araldite^®^ 2015 adhesives decreases continuously with the extension of aging time. The decrease in Tg was mainly concentrated in the early phase of aging, that is, 0–240 h of aging. The Tg of the two adhesives decreased by 14.31 °C and 27.17 °C in a pure water environment, respectively. Secondly, when the salt ion concentration increased, the Tg attenuation became smaller. The Tg decreased by 20.76 °C, 18.04 °C, and 15.82 °C in Araldite^®^ 2012 adhesive and 34.63 °C, 25.11 °C, and 22.45 °C in Araldite^®^ 2015 adhesive after 30 days of aging in DI water, 3.5% NaCl, and 5% NaCl, respectively. Compared with Araldite^®^ 2012, the Tg of Araldite^®^ 2015 was more susceptible to the aging environment. However, in a saline environment, this difference becomes blurred. The adhesive of the change tendency of Tg is similar to moisture absorption. Water entering the adhesive destroys the intermolecular hydrogen bond and weakens the interchain stress, which leads to the order of the molecular chain reduction and the increment of mobility. This change also corresponds to the trend of failure strength and further explains the change in the mechanical properties of joints.

At present, the most important method to limit the mobility of molecular chains is through the addition of different mixtures. For example, Bhagavathi [36] et al. used microcrystalline cellulose and saw chip filler to strengthen a one-component polyurethane adhesive and found that the joint of this reinforced adhesive is more resistant to high temperature and fatigue. Hasan Ulus et al. [37] uniformly distributed a type of Eloxite nanotube in the adhesive and found that the aging rate of the storage modulus of the reinforced adhesive was reduced by 43% through a dynamic mechanics test. Therefore, the addition of multiple mixtures in adhesives is becoming a new way to enhance the various properties of joints.

### 3.3. Fourier Infrared Spectrometer Analysis

In a combined environment of temperature and humidity, composite materials undergo a series of chemical reactions, such as post-curing and oxidative degradation. The FTIR test of BFRP before and after the temperature and humidity aging treatment can prove the change in its chemical composition and understand the thermal behavior mechanism that may occur during high-temperature aging.

According to the spectrogram of Araldite^®^ 2012 and Araldite^®^ 2015 adhesives in Figure 7 and Table 7, the characteristic graph of the infrared spectrum shows that the curve in the image can roughly divide into two parts, the fingerprint region of 400–1330 cm^−1^ and the characteristic frequency region of 1330–4000 cm^−1^. In addition, the benzene ring often stays constant during the reaction, so the corresponding frequency of 1508 cm^−1^ can be used as a reference [38]. According to the spectrum, the type of functional groups in the adhesive, namely, the position of the absorption peak, has little change, but the intensity of the absorption peak shows a difference. Before and after the adhesive aging, this gap is most obvious. In the spectrogram of Araldite^®^ 2012 adhesives, after the aging test, the transmittance of the hydroxyl (-OH) functional group in the infrared spectrum of the adhesive significantly decreased, while the transmittance of the ester group (-(C=O)-O) significantly increased, indicating that the ester group of the material reacts to produce hydroxyl functional group [39].

### 3.4. TGA-DTG Analysis

Araldite^®^ 2012 and Araldite^®^ 2015 are both thermosetting adhesives. The chemical chain breaks, and volatile substances are released at high temperatures, causing quality loss. Thermogravimetric analysis can be used as a method to estimate thermal stability. The thermogravimetric analysis (TGA) and the first derivative curve (DTG) of Araldite^®^ 2012 and Araldite^®^ 2015 two adhesives in three environments are shown in Figure 8, respectively. The experimental temperature is set at 40–800 °C. The curves in the figure have been marked with red and green coils and arrows to identify the x and y axes to which they belong, and T_10%_ represents the reference line for a 10 percent weight loss.

As for Araldite^®^ 2012 adhesives, the three parameters of decomposition initiation temperature, maximum weight loss rate temperature, and residue rate are used to reflect the thermal degradation process of composite materials. The intersection of the T10% line and the curve was taken as the decomposition starting temperature. The apex of the DTG curve corresponds to the inflection point of the TG curve, where the temperature is the maximum weight loss rate temperature. The residue rate is subject to 800 °C.

According to Figure 8a–c and Table 8, the thermal degradation-related data of Araldite^®^ 2012 can be obtained. It can be seen that under the three environments, the degradation onset temperature is around 345 °C, and it decreases slightly with the extension of the aging time. The maximum degradation rate is basically around 372 °C. The maximum decomposition rate of the chemical is generally greater than that of the unaged one, and there is the residue at 800 °C. All DTG curves have only one peak, indicating that degradation has occurred, which is the decomposition of epoxy resin. In thermal aging at high temperatures in the air, the methylene groups in the network may be oxidized to carbon groups, and the methylene groups are oxidized to form saturated aldehydes, ketones, or acids in other locations [40].

According to the TGA-DTG image in Figure 8d–f and the data in Table 9, information about the thermal degradation of Araldite^®^ 2015 is obtained. The maximum degradation rate occurs around 370 °C, indicating that the main degradation is also the epoxy resin matrix. However, the obvious difference is that the degradation of Araldite^®^ 2015 can be divided into three stages. In the first stage, the maximum degradation occurs at 130–165 °C, and the mass reduction is about 8–15%; in the second stage, the main degradation stage occurs at 365–370 °C, and the mass reduction is about 50%; the third stage occurs at 685–720 °C with a mass loss of about 20%.

Araldite^®^ 2012 and Araldite^®^ 2015 adhesives are both epoxy adhesives, and epoxy resin degradation occurs, so the maximum degradation is at the stage of 365–372 °C. However, due to the difference in composition between Araldite^®^ 2015 and Araldite^®^ 2012, the number of degradation stages is different.

### 3.5. Failure Strength Analysis of Adhesive Joints

The mechanical properties of Araldite^®^ 2012 and Araldite^®^ 2015 adhesive joints aged at 80 °C/DI water, 80 °C/3.5% NaCl solution, and 80 °C/5% NaCl solution were tested by quasi-static tensile tests. The mechanical property data obtained were statistically processed to obtain the variation law of the average failure strength of joints with aging time.

The average failure strength (AFS) data and fracture surface images are shown in Figure 9 and Table 10. Both Araldite^®^ 2012 and Araldite^®^ 2015 adhesives can be classified as epoxy adhesives, but due to differences between components, the AFS of Araldite^®^ 2012 adhesives were significantly higher than AFS of Araldite^®^ 2015 adhesives. With the increase in salt ion concentration in the environment, the slope of the AFS curve of Araldite^®^ 2012 tended to be gradual. After 720 h of aging, AFS remained at 10.56 MPa and 10.66 MPa when NaCl concentration was 3.5% and 5%, respectively. Meanwhile, the AFS of Araldite^®^ 2015 in a salt ion environment is generally higher than that of deionized water.

The main influencing factors of AFS at constant 80 °C are aging time and ion content in solution. Under the three conditions, the AFS of all joints decreases with the increase in time. In a deionized water environment, Araldite^®^ 2012 and Araldite^®^ 2015’s AFS loss is mainly concentrated in 0–240 h and 240–480 h, and the decrease in these adhesives in the above two periods were 22.47%, 12.57%, and 16.27%, 18.25%, respectively. When the aging environment contains salt ions, the average failure intensity loss is concentrated in 240–480 h. Their AFS decreased by 9.08% and 19.56% in 3.5% NaCl solution, respectively. In the 5% NaCl solution, the decreases were 6.29% and 22.39%, respectively. The curves between the two NaCl solutions are relatively close because of the small difference in ion concentration.

Since the specimen is immersed in deionized water and NaCl solution, water can have a negative influence on the failure strength. Polymers and water molecules belong to polar molecules, which will interact under an aging environment, leading to changes in the physical, chemical, and mechanical properties of materials. In general, water may degrade BFRP composites by one or a combination of the following: (I). Changing the resin matrix; (II). Damage of fiber/matrix interface; (III). Fiber level degradation [41]. In Araldite^®^ 2012 and Araldite^®^ 2015 epoxy adhesives, water will increase the ductility of resin/adhesive and reduce the elastic modulus and strength. Water exists in epoxy adhesive as type I bound water and type II bound water. Type I bound water acts as a plasticizer to destroy the van der Waals force between chains. Type II bound water is the product of a strong hydrogen bond between water and resin network, which is greater than the adhesive force between adhesive and composite material, causing irreversible damage [42]. Therefore, the impression of water on adhesive joints can be divided into the following three forms [43]: (I). Changes in the properties of the adhesive body; (II). Changes in the properties of bonding materials; (III). Changes in the properties of the adhesive joint interface.

Figure 10 shows the damage to the polymer chain by moisture. The molecular chain is shown in blue, the yellow part indicates the beginning of the chain break, and the red part indicates the extension of the chain break. When water molecules do not enter the molecular chains, hydrogen bonds can be formed between the molecular chains according to the polar groups, thus reaching a very stable state. Then, because of the erosion of the environment, water molecules begin to enter between the molecular chains. In the end, the polarity of water molecules is stronger than that of molecular chain groups, which leads to the formation of hydrogen bonds between water molecules and groups on the molecular chain, thus destroying the hydrogen bonds between molecules, leading to the reduction in the order of molecular chains and the increase in mobility [44].

For joints using epoxy resin adhesives and FRP composite materials, the aging mechanism is controlled by the moisture diffusion of polymer materials, adhesives, and fiber/matrix interface. Under the competition between the degradation of surface coating in composite materials and the debonding damage mechanism of the aluminum/epoxy interface caused by adhesive expansion, different types of failure joints appear [45]. At Araldite^®^ 2012 failure interface, the interface failure area between the adhesive and 5052 Al gradually decreases with the aging time, and changes to tear failure between BFRP and adhesive. In salt ion solution, cohesion failure occurs in the part of the interface after 240 and 720 h of aging. As for Araldite^®^ 2015 failure interface, the interface failure of adhesive and 5052 Al coexists with the tear failure of the BFRP board. It is worth noting that a thin layer of adhesive is distributed in some BFRP bonding areas, called thin-layer cohesion failure. After the aluminum surface is effectively treated, the polymer will penetrate the pits and holes on the metal surface, and the bonding strength is also improved through mechanical interlocking, leading to the failure that may be transferred to the adhesive close to the interface [46,47].

BFRP and 5052 Al can be approximated as rigid bodies, so the displacement measured by the tensile machine can be attributed to the displacement generated in the bonding area. The most representative load-displacement curves of different aging stages are shown in Figure 11. The maximum failure strength of the joint is shown in Table 11.

The maximum loading of Araldite^®^ 2012 adhesives was significantly higher than that of Araldite^®^ 2015 adhesives; at the same time, Araldite^®^ 2012 adhesive joints can withstand greater tensile displacement. The curves of all specimens also have similar points. First, the slope of the load-displacement curve of all specimens decreased gradually. As the load increases, the resistance to deformation decreases, so the stiffness decreases and the slope of the curve also decreases. Second, the stiffness decrease of all specimens was concentrated in 0–240 h. When the experiment began, the temperature jumped from room temperature to 80 °C. At high temperatures (80 °C), the joint shows a greater degree of strain rate sensitivity, and the joint can withstand significantly lower than that at room temperature [48]. These differences are attributed to the softening of the adhesive at high temperatures. The resin matrix and adhesive will soften under high temperatures, leading to an increase in viscoelastic reaction and a significant decrease in joint stiffness. Therefore, the curve slope is large when not aged and small after aging. During aging, with the increase in time, the water in the epoxy resin becomes a plasticizer, reducing joint stiffness.

A scanning electron microscope (SEM3300, CIQTEK, Hefei, Anhui Province, China) was used, and a small area to observe the joint failure position was selected. SEM figures are shown in Figure 12. The green dashed line is the direction of fiber orientation. The image in the red dashed line box is the image after the resin and fiber are debonded and pulled out. The image in the blue dashed line box is the image of multiple damage types of fiber and resin. 

Debonding at the interface between resin and fiber is observed in the diagram. Fibers leave complete grooves at the resin/fiber interface after drawing out, and the concomitant presence of resin crack, resin debris, resin burr, and resin void. The Araldite^®^ 2012 (3.5% NaCl/720 h) and Araldite^®^ 2015(DI water/720 h) images show intact fiber bundles exposed to the surface. The Young’s modulus of basalt fiber ranges from 93.1 to 110 GPa, and the Young’s modulus of resin matrix was 33.6 GPa [49]. By comparison, the resin matrix is more vulnerable to damage, which also explains that the fiber bundle is relatively intact while the matrix breaks and leaves debris. In conclusion, the failure of the composite plate occurs at the interface between fiber and resin.

## 4. Conclusions

In terms of average failure strength, the initial average failure strength of Araldite^®^ 2012 adhesives is relatively higher than Araldite^®^ 2015 adhesives. Within 0–480 h, the destruction strength of the two adhesives decreases significantly. In the salt solution environment, the average failure strength of the two adhesives increases, and the period of maximum decline in failure strength is delayed, from 0 to 480 h. As for stiffness, Araldite^®^ 2012 can withstand larger loads and displacements than Araldite^®^ 2015, and the stiffness attenuation of the two epoxy adhesives mainly occurs from 0–240 h. In terms of failure image, with the aging time prolongation, the Araldite^®^ 2012 adhesive joint changes from interface failure to tear failure, while the Araldite^®^ 2015 adhesive joint has both interface failure and tear failure during aging, and there is still a thin layer of adhesive on the part of the tear surface. Scanning electron microscopy (SEM) images of both Araldite^®^ 2012 and Araldite^®^ 2015 joints showed fiber withdrawal and resin damage due to tensile destruction.In the water absorption test, it can be seen that the moisture absorption of Araldite^®^ 2015 bulk adhesive is higher than that of Araldite^®^ 2012, and moisture absorption periods are also concentrated in 0–240 h.From the test characterization, FTIR images show that the two adhesives belong to epoxy adhesives, so the group absorption peaks are similar, and both have hydrolysis of ester groups. However, due to the differences in composition, absorption peaks are still different. The absorption peaks of the two adhesives decrease with the aging time. As can be seen from TGA-DTG images, both Araldite^®^ 2012 and Araldite^®^ 2015 have a large degree of thermogravimetric loss near 360 °C, but Araldite^®^ 2012 has only one degradation stage, while Araldite^®^ 2015 has three, which also indicates the difference in composition.

## Figures and Tables

**Figure 1 polymers-15-03892-f001:**
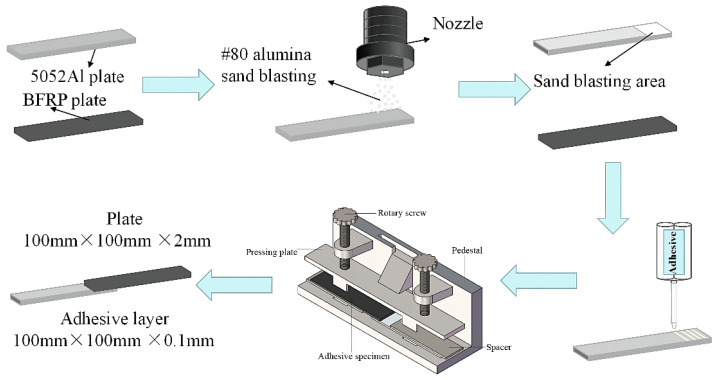
The test specimen fabrication process.

**Figure 2 polymers-15-03892-f002:**
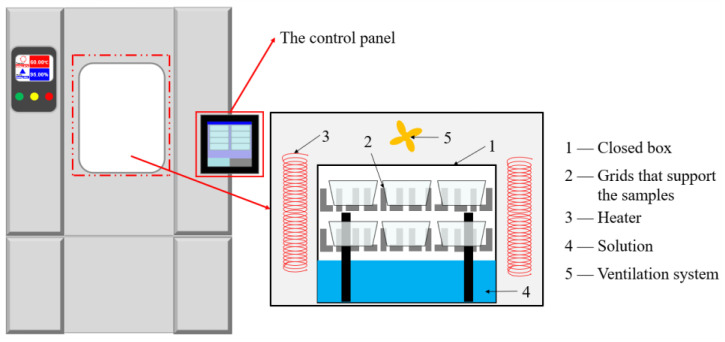
High-low temperature damp-heat alternating experiment box.

**Figure 3 polymers-15-03892-f003:**
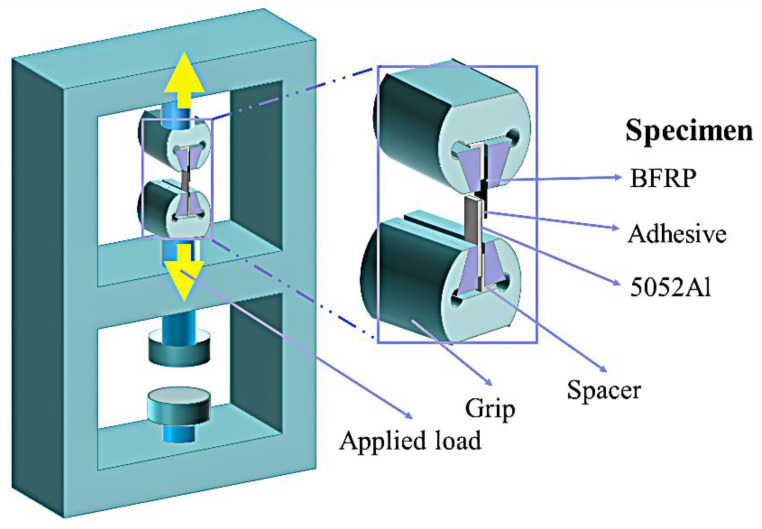
Quasi-static tensile test.

**Figure 4 polymers-15-03892-f004:**
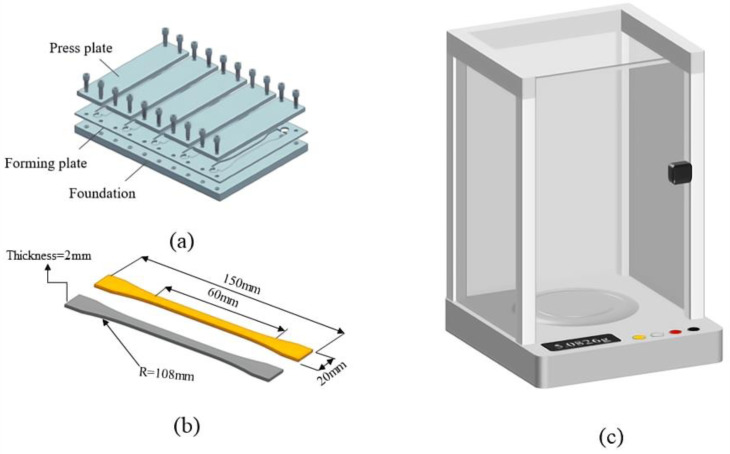
Water absorption test instrument and sample: (**a**) Dog-bone sample making mold; (**b**) Dog-bone sample; (**c**) Analytical balance.

**Figure 5 polymers-15-03892-f005:**
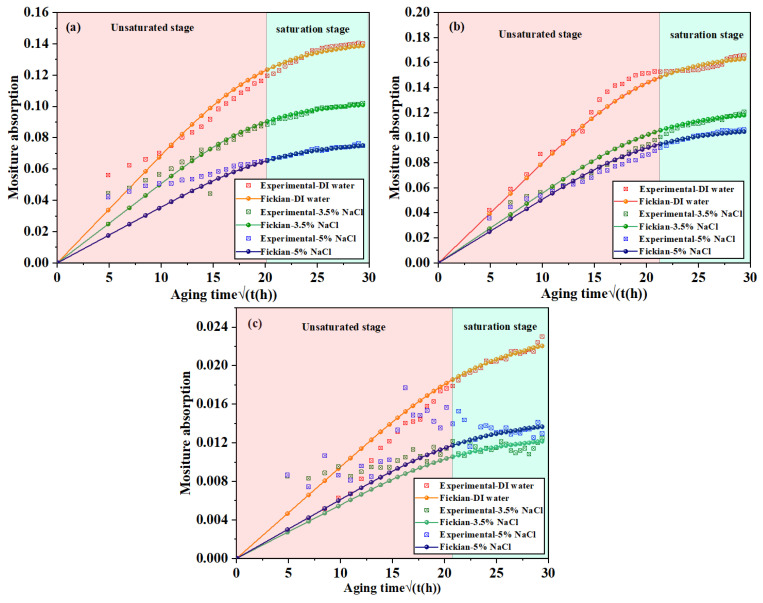
Moisture absorption and curve fitting of three different composites under three different environments: (**a**) Araldite^®^ 2012, (**b**) Araldite^®^ 2015, and (**c**) BFRP.

**Figure 6 polymers-15-03892-f006:**
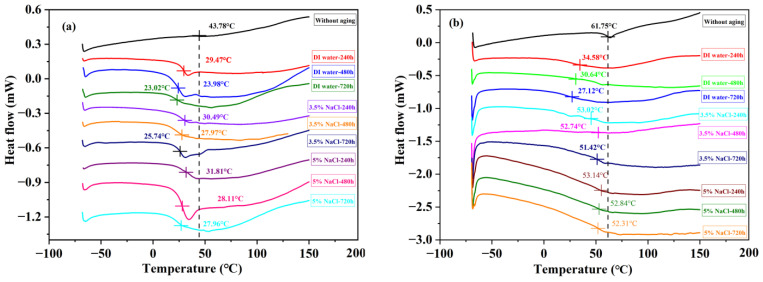
DSC thermogram: (**a**) Araldite^®^ 2012 and (**b**) Araldite^®^ 2015.

**Figure 7 polymers-15-03892-f007:**
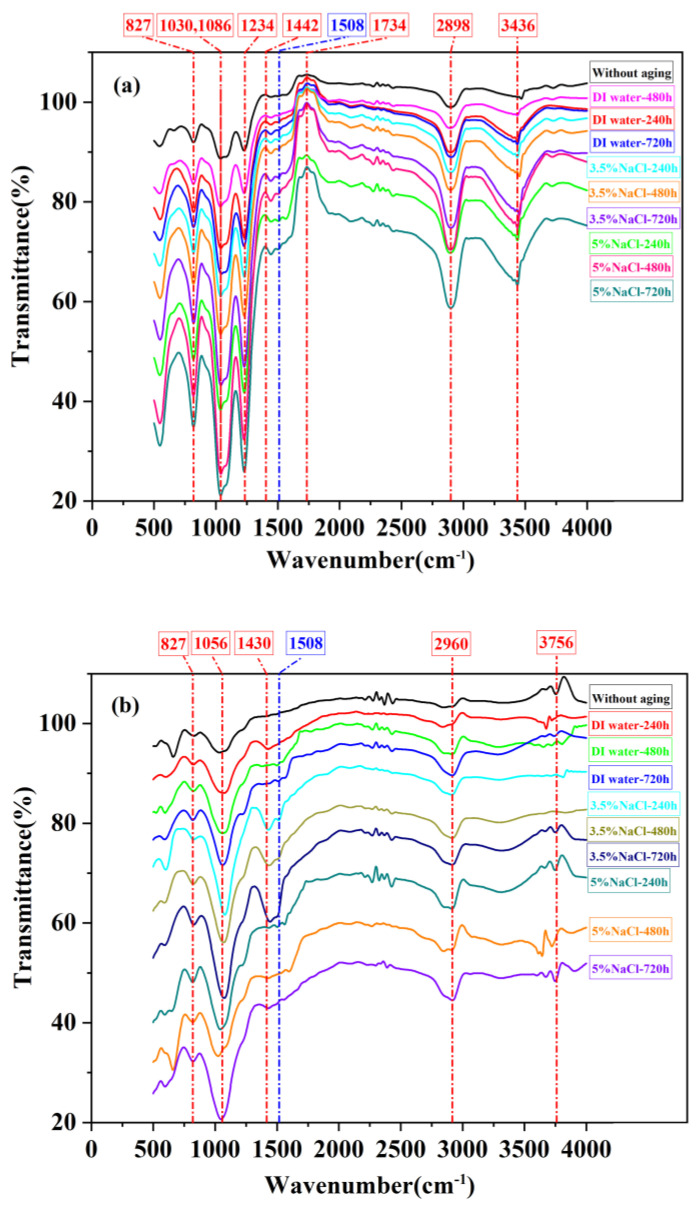
FTIR spectra of Adhesive: (**a**) Araldite^®^ 2012 and (**b**) Araldite^®^ 2015.

**Figure 8 polymers-15-03892-f008:**
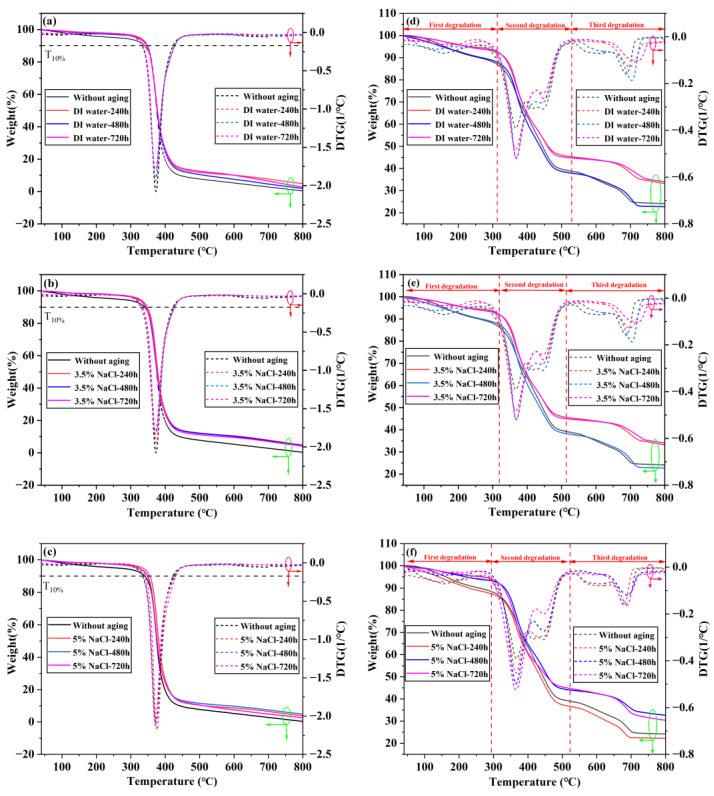
TGA-DTG images of adhesive joint: (**a**) Araldite^®^ 2012 joints 80 °C/DI water, (**b**) Araldite^®^ 2012 joints 80 °C/3.5% NaCl, (**c**) Araldite^®^ 2012 joints 80 °C/5% NaCl, (**d**) Araldite^®^ 2015 joints 80 °C/DI water, (**e**) Araldite^®^ 2015 joints 80 °C/3.5% NaCl, and (**f**) Araldite^®^ 2015 joints 80 °C/5% NaCl.

**Figure 9 polymers-15-03892-f009:**
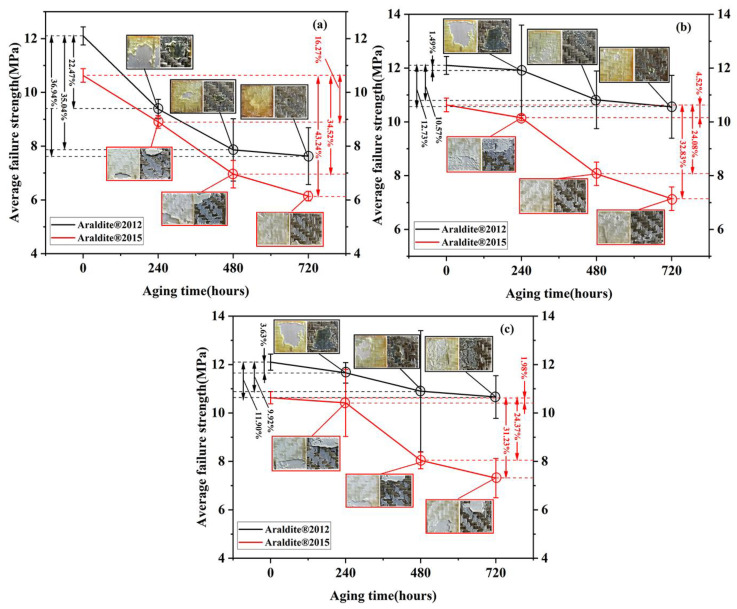
Failure load curve and failure section picture of Araldite^®^ 2012 and Araldite^®^ 2015: (**a**) 80 °C/DI water, (**b**) 80 °C/3.5% NaCl, and(**c**) 80 °C/5% NaCl.

**Figure 10 polymers-15-03892-f010:**
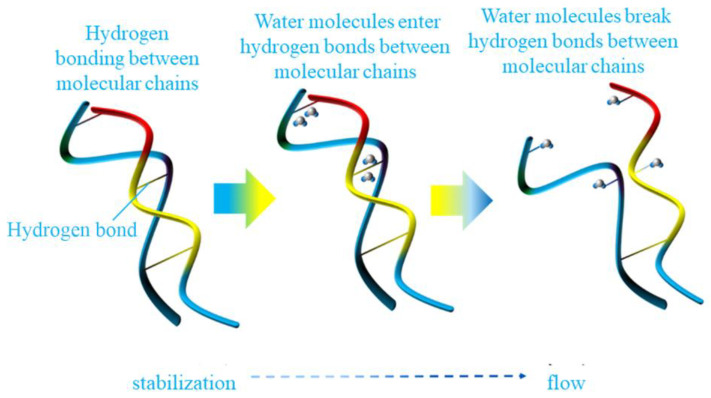
The damage of polymer chain by moisture.

**Figure 11 polymers-15-03892-f011:**
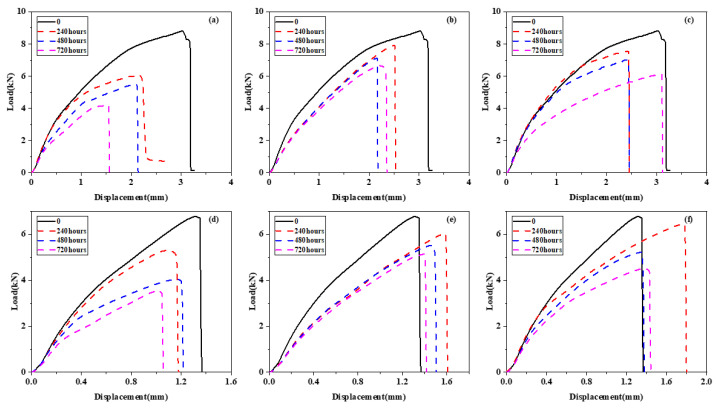
The load-displacement curve of adhesive joint: (**a**) Araldite^®^ 2012 joints 80 °C/DI water, (**b**) Araldite^®^ 2012 joints 80 °C/3.5% NaCl, (**c**) Araldite^®^ 2012 joints 80 °C/5% NaCl, (**d**) Araldite^®^ 2015 joints 80 °C/DI water, (**e**) Araldite^®^ 2015 joints 80 °C/3.5% NaCl, and (**f**) Araldite^®^ 2015 joints 80 °C/5% NaCl.

**Figure 12 polymers-15-03892-f012:**
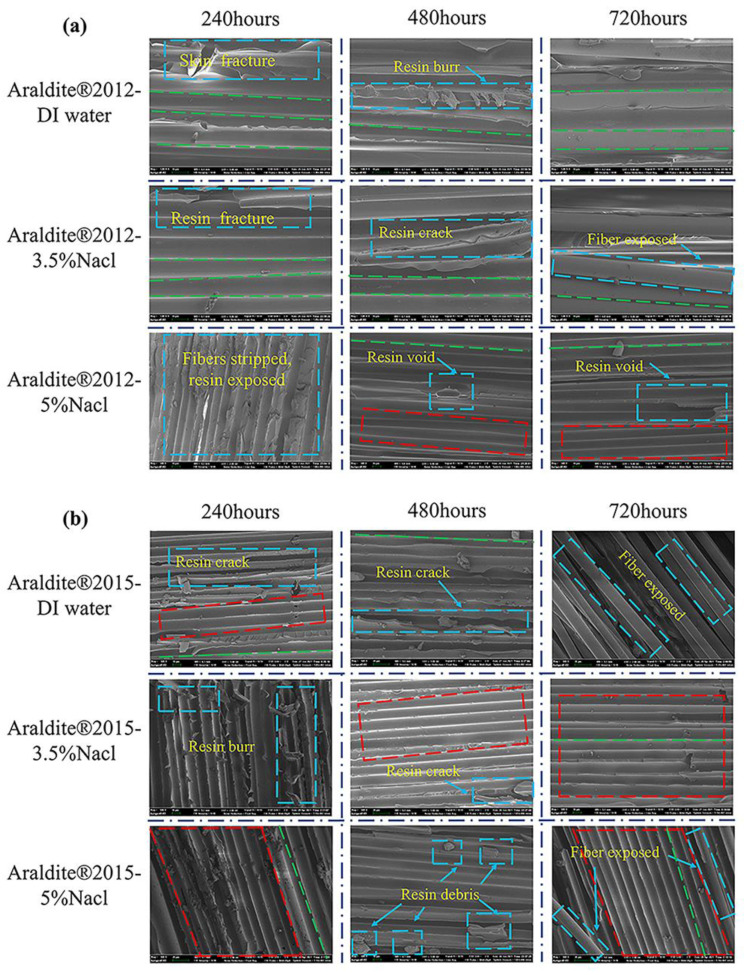
SEM images: (**a**) Araldite^®^ 2012 and (**b**) Araldite^®^ 2015.

**Table 1 polymers-15-03892-t001:** Material properties of basalt fiber unidirectional fabric.

GT-807A/GT807B Epoxy Resin		Basalt Fiber Unidirectional Fabric	
Cure condition	25 °C × 24 h + 90 °C × 3 h	Density/(g/cm^2^)	0.00265
Density/(kg/m^3^)	0.0015	Tensile strength/(MPa)	2100
Tensile strength/(MPa)	75.5	Young’s modulus/(GPa)	105
Tensile modulus/(MPa)	3282.5	Nominal thickness/(mm)	0.115
Tg/(°C)	≥80	Single fiber size/(μm)	13

**Table 2 polymers-15-03892-t002:** The properties of the 5052 aluminum alloy material.

Mechanical Properties	Numerical Value
Density/(kg/m^3^)	2730
Young’s modulus/(GPa)	70
Poisson’s radio	0.33
Yield strength/MPa	227
Tensile strength/MPa	378

**Table 3 polymers-15-03892-t003:** Araldite^®^ 2012 and Araldite^®^ 2015 adhesives parameters.

	Araldite^®^ 2012	Araldite^®^ 2015
Young’s modulus/E(GPa)	1.65	1.85
Shear modulus/G(GPa)	0.25	0.56
Density/(kg/m^3^)	1.18	1.60
Poisson’s radio	0.43	0.33

**Table 4 polymers-15-03892-t004:** Araldite^®^ 2012 and Araldite^®^ 2015 Moisture uptake of three different environments.

	Araldite^®^ 2012		Araldite^®^ 2015		Thickness (T = 2 h, mm)
	Saturation Moisture Uptake M_∞_ (%)	Diffusion Coefficient D × 10^−3^ (mm^2^/s)	Saturation Moisture Uptake M_∞_ (%)	Diffusion Coefficient D × 10^−3^ (mm^2^/s)	
DI Water	14.05	1.89	16.57	1.83	2
3.5% NaCl	10.22	1.94	12.07	1.68	2
5% NaCl	7.49	1.79	10.66	1.79	2

**Table 5 polymers-15-03892-t005:** BFRP Moisture uptake of three different environments.

Sample (BFRP)	Saturation Moisture Uptake M_∞_(%)	Diffusion Coefficient D × 10^−3^ (mm^2^/s)	Thickness (T = 2 h, mm)
DI Water	2.30	1.34	2
3.5% NaCl	1.25	1.55	2
5% NaCl	1.41	1.48	2

**Table 6 polymers-15-03892-t006:** The glass transition temperature of Araldite^®^ 2012 and Araldite^®^ 2015.

	Glass Transition Temperature(Tg/°C)					
	Araldite^®^ 2012			Araldite^®^ 2015		
Time (h)	DI Water	3.5% NaCl	5% NaCl	DI Water	3.5% NaCl	5% NaCl
0	43.78			61.75		
240	29.47	30.49	31.81	34.58	42.53	46.25
480	23.98	27.97	28.11	30.64	40.38	42.48
720	23.02	25.74	27.96	27.12	36.64	39.30

**Table 7 polymers-15-03892-t007:** Araldite^®^ 2012 and Araldite^®^ 2015 wavenumber of functional group.

Araldite^®^ 2012		Araldite^®^ 2015	
Wavenumber/cm^−1^	Functional Group	Wavenumber/cm^−1^	Functional Group
827	p-phenylene groups	827	p-phenylene groups
1030, 1086	Stretching of the trans forms of the ether linkage	1056	Stretching of the trans forms of the ether linkage
1234	Stretching mode for aromatic ether	1430	C-H bending vibration of the ester group
1442	C-H bending vibration of the ester group	2960	-CH3 sway deformation vibration vibration
1726	ester group-(C=O)-O	3756	-OH asymmetrical stretching
2898	-CH3 symmetric stretching		
3436	-OH symmetric stretching		

**Table 8 polymers-15-03892-t008:** TGA-DTG data of Araldite^®^ 2012.

Environment	Aging Time (h)	Initial Pyrolysis Temperature (°C)	Maximum Weight Loss Rate Temperature (°C)	Residue Rate (%)
Without aging	0	340.0	373.3	0
DI water	240	348.7	372.5	4.89%
	480	346.2	371.7	1.88%
	720	345.7	371.7	2.68%
3.5% NaCl	240	347.5	374.2	4.82%
	480	343.3	371.2	4.43%
	720	344.2	372.5	4.15%
5% NaCl	240	355.3	375.8	3.79%
	480	348.8	371.7	4.82%
	720	347.9	371.7	2.56%

**Table 9 polymers-15-03892-t009:** TGA-DTG data of Araldite^®^ 2015 adhesives.

Environment	Aging Time (h)	First Degration’s Temperature (°C)	Second Degration’s Temperature (°C)	Third Degration’s Temperature (°C)
Without aging	0	157.5	367.5	688.3
DI water	240	165.0	366.7	697.5
	480	135.4	369.2	704.2
	720	150.0	368.3	718.8
3.5% NaCl	240	165.0	366.7	679.5
	480	134.5	368.3	704.2
	720	150.0	368.3	718.7
5% NaCl	240	136.7	365.8	678.3
	480	142.8	365.0	689.2
	720	146.7	365.8	685.8

**Table 10 polymers-15-03892-t010:** Average failure strength data.

	Average Failure Strength (MPa)					
	Araldite^®^ 2012			Araldite^®^ 2015		
Time (h)	DI Water	3.5% NaCl	5% NaCl	DI Water	3.5% NaCl	5% NaCl
0	12.10	12.10	12.10	10.63	10.63	10.63
240	9.38	11.92	11.66	8.90	10.15	10.42
480	7.86	10.82	10.90	6.96	8.07	8.04
720	7.63	10.56	10.66	6.14	7.14	7.31

**Table 11 polymers-15-03892-t011:** Statistical table of adhesive failure strength.

	Failure Load (kN)					
	Araldite^®^ 2012			Araldite^®^ 2015		
Time (h)	DI Water	3.5% NaCl	5% NaCl	DI Water	3.5% NaCl	5% NaCl
0	8.814	8.814	8.814	6.775	6.775	6.775
240	6.053	7.894	7.550	5.293	6.040	6.450
480	5.484	7.104	7.016	4.026	5.513	5.221
720	4.186	6.648	6.106	3.532	5.137	4.506

## Data Availability

Data will be made available on request.

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
