# Peer review of "Comparison of Mechanical Properties of Ductile/Brittle Epoxy Resin BFRP-AL Joints under Different Immersion Solutions"

_polymers, 2023, doi:10.3390/polym15193892_

Round 1

Reviewer 1 Report

Due to the issues of global warming, environmental contamination and energy crisis are becoming more and more serious; energy conservation and emission reduction are a significant subject for sustainable development of automotive, railway vehicle and aircraft industries. Therefore, the development of  joining processes suitable for use with lightweight materials, such as aluminium and high-performance fiber-reinforced polymer composite material, is attracting  attention. The reviewed paper (manuscript ID: polymers- 2602516, titled:" Comparison of mechanical properties of ductile/brittle epoxy  resin BFRP-AL joints under different immersion solutions“) is well done. This manuscript presents the original results of research on the properties of bonded joints of materials -  basalt fiber reinforced  polymer composite and aluminum alloy 5052 - before and after aging in various environments. I have no objections against the work by essence, but in my opinion:

1. Please consider reviewing the abstract and highlighting novelty. Please use numbers or % terms to clearly shows us the results in your work.  

2. Make correction: Density units should be given kg/m3.

3. Line 164 – Make correction: aging time units /hours/ are missing.

4. Subchapter 2.3.2. - Make correction: Add the standard according to which the test was done.

5. All equipment used should be fully defined i.e., type, manufacturer, country (Subchapter 2.3.3.- Analytical balance, Subchapter 2.3.4.- DSC Q100, Subchapter 2.3.5. - infrared spectrometer., Line 466 - SEM)

6.  Line 232 - Make correction: Fick fitting curve are shown in Figure 5.

7. Figure 6, page 9 - Make correction: The description of the curves in Figure 6 is difficult to read.

8. Line 345 - Make correction: According to Fig. 8a, Fig. 8b, Fig. 8c, ...

9. Line 357 -  Make correction: According to the TGA-DTG image in Fig. 8d, Fig. 8e, Fig. 8f …

10. References must be prepared according to the rules for MDPI Journals -  Add Digital Object identifier (DOI).

11. In general, what is the reproducibility of those experiments?

Author Response

1. Summary

Thank you very much for taking the time to review this manuscript. Those comments are all valuable and very helpful for revising and improving our paper, as well as the important guiding significance to our researches. We have studied comments carefully and have made correction which we hope meet with approval. Please find the detailed responses below and the corresponding revisions in the re-submitted files.

Thank you very much for taking the time to review this manuscript.

2. Questions for General Evaluation

Reviewer’s Evaluation

Response and Revisions

Does the introduction provide sufficient background and include all relevant references?

Yes/Can be improved/Must be improved/Not applicable

Thank you for your comments.

Are all the cited references relevant to the research?

Yes/Can be improved/Must be improved/Not applicable

Thanks for your comments, we have adjusted the references

Is the research design appropriate?

Yes/Can be improved/Must be improved/Not applicable

Thank you for your comments.

Are the methods adequately described?

Yes/Can be improved/Must be improved/Not applicable

Thank you for your comments. We will adjust the criteria for method need in the comments.

Are the results clearly presented?

Yes/Can be improved/Must be improved/Not applicable

Thanks for your comments, we have revised the analysis of the results of unclear expression.

Are the conclusions supported by the results?

Yes/Can be improved/Must be improved/Not applicable

Thank you for your comments.

3. Point-by-point response to Comments and Suggestions for Authors

Comments 1: Please consider reviewing the abstract and highlighting novelty.  Please use numbers or % terms to clearly shows us the results in your work.

Response 1:  Thanks for your advice, We have improved the summary section based on your suggestions, please refer to Page 1. Thank you again.

Comments 2: Make correction: Density units should be given kg/m3.

Response 2: Thanks for your advice, we have adjusted the density units to kg/m3, please refer to Table 1 and 3. Thank you again.

Comments 3: Line 164 – Make correction: aging time units /hours/ are missing.

Response 3: Thank you for pointing this out, we have replaced the missing units in their respective positions, please refer to line 162. Thank you again.

Comments 4: Subchapter 2.3.2. - Make correction: Add the standard according to which the test was done.

Response 4: We thank the reviewer for points this out. We have added the reference standard ASTM D 5868-01 to line 174, and updated the references. Thank you again.

Comments 5: All equipment used should be fully defined i.e., type, manufacturer, country (Subchapter 2.3.3.- Analytical balance, Subchapter 2.3.4.- DSC Q100, Subchapter 2.3.5. - infrared spectrometer., Line 466 - SEM) 

Response 5: Yes we fully agree with the reviewer that we have fully defined equipment, please refer to line 200, 211 and 467. Thank you again.

Comments 6:  Line 232 - Make correction: Fick fitting curve are shown in Figure 5.

Response 6: Thanks for your suggestion, we are sorry to neglect this mistake, and we have changed the number, please refer to line 232. Thank you again.

Comments 7: Figure 6, page 9 - Make correction: The description of the curves in Figure 6 is difficult to read.

Response 7: Thanks for your suggestion, We have modified the analysis of the curves in Figure 6, please refer to page 9. Thank you again.

Comments 8: Line 345 - Make correction: According to Fig. 8a, Fig. 8b, Fig. 8c, ...

Response 8: We feel sorry for the inconvenience brought to the reviewer, the correct number have written in paper, please refer to line 348.      

Comments 9: Line 357 - Make correction: According to the TGA-DTG image in Fig. 8d, Fig. 8e, Fig. 8f …

Response 9: Thank for your careful check, the correct number have written in paper, please refer to line 360.     

Comments 10: References must be prepared according to the rules for MDPI Journals - Add Digital Object identifier (DOI).

Response10: Thank for your rigorous advice, we have changed our references and  have added the doi. Please refer to page 19 and 20. Thank you again.

Comments 11: In general, what is the reproducibility of those experiments?

Response11: Thanks for your question, as for experiment reproducibility, we would like to answer from the following aspects: (1). There is a unified process for making test samples. The cutting and surface treatment of the substrate have corresponding operating specifications, and the temperature and humidity are also accurately controlled during production; (2). The aging stage of the specimen is carried out by a high-low temperature damp-heat alternating experiment box. The instrument runs stably and controls the aging temperature and humidity accurately; (3). The subsequent testing of specimens also complies with the corresponding standards and test specifications. Thus, we can show that our experiments are reproducible. About this information, please refer to pages 3-6 for this information. Thank you again.

4. Response to Comments on the Quality of English Language

Point 1: I am not qualified to assess the quality of English in this paper

Response 1: Thank you for your assessment of our English language.

5. Additional clarifications

Thank you, we have no additional clarifications.

Reviewer 2 Report

This article investigates an important topic of fiber-reinforced polymers. However, it is written in a very confusing manner, the following must be resolved:

1. The clear sample labeling and the precise concentration of fibers and epoxy must be included

2. Thermal analysis of samples shows no influence of fibers, only adhesives are presented; The same is confusing for FTIR

3. Authors from this group have published quite a similar study in Polymers this year. State the novelty compared to this study and quote the published research:

https://doi.org/10.3390/polym15153232

Author Response

1. Summary

Thank you very much for taking the time to review this manuscript. Those comments are all valuable and very helpful for revising and improving our paper, as well as the important guiding significance to our researches. We have studied comments carefully and have made correction which we hope meet with approval. Please find the detailed responses below and the corresponding revisions in the re-submitted files.

Thank you very much for taking the time to review this manuscript.

2. Questions for General Evaluation

Reviewer’s Evaluation

Response and Revisions

Does the introduction provide sufficient background and include all relevant references?

Yes/Can be improved/Must be improved/Not applicable

Thanks for your comments and we have cited previous studies

Are all the cited references relevant to the research?

Yes/Can be improved/Must be improved/Not applicable

Thanks for your comments.

Is the research design appropriate?

Yes/Can be improved/Must be improved/Not applicable

Thanks for your comments, we have optimized the material parameters.

Are the methods adequately described?

Yes/Can be improved/Must be improved/Not applicable

Thank you for your comments. We will adjust the criteria for method need in the comments.

Are the results clearly presented?

Yes/Can be improved/Must be improved/Not applicable

Thanks for your comments, we have revised the analysis of the results of unclear expression.

Are the conclusions supported by the results?

Yes/Can be improved/Must be improved/Not applicable

Thanks for your comments.

3. Point-by-point response to Comments and Suggestions for Authors

Comments 1: Please consider reviewing the abstract and highlighting novelty. Please use numbers or % terms to clearly shows us the results in your work.  

Response 1: Thank you for pointing this out. We have revised the analysis of the results of unclear expression. We have updated the parameters of fiber and resin in BFRP in the table and determined that the volume yarn content of BFRP is 65.18%. Please refer to Table 1 and line 122. Thank you again.

Comments 2: Thermal analysis of samples shows no influence of fibers, only adhesives are presented; The same is confusing for FTIR.

Response 2: Thank you for point it out. According to the literature named Environmental durability of adhesively bonded FRP/steel joints in civil engineering applications: State of the art, adhesives are the weak part of composite and metal bonding components, the effectiveness and success of the FRP/metal joint is dependent on the quality, integrity and durability of the adhesive bond between the adherends. In section analysis, it is found that adhesive layer damage is the main failure mode, and gradually becomes cohesive failure with aging time increasing. Therefore, we choose adhesive as the sample for thermal analysis, and we have added explanation in the paper, please refer to line 278. Thank you again.

Comments 3: Authors from this group have published quite a similar study in Polymers this year. State the novelty compared to this study and quote the published research.

Response 3: We totally understand the reviewer’s concern. With regard to this question, we will answer it from the following aspects:(1). The previous study has primarily concentrated on the bonding characteristics of the identical composites both pre- and post-aging. On this basis, this study focuses on the bonding of dissimilar substrates; (2). The aging environment of the previous study was salt spray, but this study added pure water environment and paid attention to the influence of salt ions on the performance of the joint; (3). In this study, FTIR tests were added to focus on the changes of functional groups before and after aging, which were also not available in previous studies. Thanks again for your suggestion, we refer to the previous study on line 180-110.

4. Response to Comments on the Quality of English Language

Point 1: English language fine. No issues detected.

Response 1: Thank you for your comments on our English expressions.

5. Additional clarifications

Thank you, we have no additional clarifications.

Round 2

Reviewer 2 Report

The authors have corrected everything in accordance with reviewer's suggestions. However, they should correct point 3. 'From the microscopic analysis' must be corrected, FTIR and TGA are not microscopic analyses.

Author Response

Thank you for reviewing our paper again. We have responded to your comments point by point. Please see the attachment. Thank you again.
